# Antimicrobial resistance and genome characteristics of *Salmonella enteritidis* from Huzhou, China

**Wei Yan, Deshun Xu** *, **Liping Chen, Xiaofang Wu**

Huzhou Center for Disease Control and Prevention, Huzhou, China

* xds666092@126.com

## Abstract

*Salmonella enteritidis* is a main pathogen responsible for sporadic outbreaks of gastroenteritis, and therefore is an important public health problem. This study investigated the drug resistance and genomic characteristics of *S. enteritidis* isolated from clinical and food sources in Huzhou, Zhejiang Province, China, from February 1, 2021, to December 30, 2023. In total, 43 *S. enteritidis* strains isolated during the study period were subjected to virulence gene, drug resistance gene, genetic correlation, antibiotic resistance, and multilocus sequence typing analyses. All 43 isolates were identified as ST11, and contained 108 virulence-related genes. Drug sensitivity analysis of the 43 isolates showed resistance rates of 100% to nalidixic acid and 90.70% to ampicillin and ampicillin/sulbactam. Multidrug resistance is a serious issue, with 81.40% of strains resistant to three or more antibacterial drugs. Genome sequencing indicated that *S. enteritidis* possessed 23 drug resistance genes, of which 14 were common to all 43 isolates. Phylogenetic analysis based on core genome single-nucleotide polymorphisms divided the 43 *S. enteritidis* strains into three clusters, with the 10 samples from an outbreak forming an independent branch located in cluster 3.

## Introduction

*Salmonella* is a zoonotic pathogen, an important pathogenic factor responsible for foodborne diseases [1, 2]; it is the main cause of bacterial food poisoning [3]. Contaminated food and water or individuals infected with *Salmonella* can become the source of infection, posing a serious threat to the health humans and other animals [4]. More than 2600 serotypes or variants of *Salmonella* have been identified to date [5, 6], of which about 50 are closely related to human health. The clinical manifestations of *Salmonella* infection are complex and diverse, and can be divided into gastroenteritis, septicemia (typhoid), and local infection types [7]. Recent studies [8] have shown that the incidences of typhoid fever and paratyphoid fever have decreased, while some non-typhoidal *Salmonella* infections are showing upward trends. Non-typhoidal *Salmonella* is frequently associated with diarrheal illness or self-limiting gastroenteritis in humans [9, 10], and causes up to 80.3 million cases of gastrointestinal disease and 150 000 deaths each year around the world [11]. In China, 70–80% of cases of foodborne disease are caused by non-typhoidal *Salmonella* [12]. *Salmonella enteritidis* is the main *Salmonella*

**Funding:** "This work was supported by grants from Huzhou Science and Technology Bureau (Award

Number: 2023GY08), the funders had no role in study design, data collection and analysis, decision to publish, or preparation of the manuscript."

serotype, accounting for 40–60% of laboratory-confirmed *Salmonella* infections in recent years in various regions of the world [13, 14].

Antimicrobial resistance remains a serious challenge in the treatment and control of *Salmonella enterica* infections. Multidrug-resistant strains are linked to more severe disease outcomes [15] and can be passed along at various points from food production to consumption [16]. With the widespread use of antibiotics, the rate of drug-resistant *Salmonella* is also increasing, that bring great difficulty for clinical treatment, necessitating the study of drug resistance in these important foodborne pathogens [17, 18].

Whole-genome sequencing (WGS) has become an economically viable alternative to conventional typing methods for investigating disease outbreaks and public health surveillance [19]. Comparative genomics with WGS provides insight into the genomes of pathogenic bacteria, including identification of candidate drug compounds, potential virulence determinants, mechanisms of pathogenicity, and the evolution of pathogens. This technology has been instrumental in improving diagnostics and public health clinical microbiology [20], and is widely used to investigate outbreaks of pathogenic bacteria, including *Bacillus cereus*, *Escherichia coli*, *Vibrio parahaemolyticus*, and *Salmonella* species [21–24].

*S. enteritidis* is the most frequently isolated serovar from food sources patients with diarrhea in Huzhou, Zhejiang Province, China [25, 26]. As far as we know, there have been few previous reports on the drug resistance and genetic characteristics of *S. enteritidis* in the Huzhou area. In this study, we sequenced the whole genomes of *S. enteritidis* isolates from diarrhea patients in Huzhou. This study aims to explore the genetic type, distribution characteristics of resistance genes and virulence genes, as well as the evolutionary relationships of strains. The findings of this research can offer valuable insights for conducting food safety risk assessments, prevention and control measures against *S. enteritidis* in Huzhou, China.

## Materials and methods

### Ethics statement

This study was approved by the Human Research Ethics Committee of the Huzhou Center for Disease Control and Prevention (approval number: HZ2020007). The only human materials used in this study were stool samples obtained from patients for routine assessment. Each participant provided oral informed consent.

### Bacterial isolates

In total, 41 *S. enteritidis* strains isolated from diarrhea patients from six active foodborne surveillance sentinel hospitals in Huzhou, Zhejiang Province, China, from February 1, 2021, to December 30, 2023 (9, 18, and 14 strains in 2021, 2022, and 2023, respectively), as well as two *S. enteritidis* strains isolated from food in November 2023 were examined. (S1 Table). Bacteria were isolated according to the methods described in *Diagnostic Criteria for Infectious Diarrhea* (WS271–2007) [27] and *National Standards for Food Safety* (GB4789.4–2016) [28]. All 43 strains of *S. enteritidis* were stored at –80˚C in porcelain culture storage tubes (Qingdao Haibo, Qingdao, Shandong, China).

### Antimicrobial susceptibility testing (AST)

The antimicrobial susceptibilities of the 43 isolates were tested using the broth microdilution method and strains were classified as sensitive, intermediate, or resistant according to the

Clinical and Laboratory Standards Institute (CLSI) breakpoints for *Salmonella* strains. *E. coli* ATCC 25922 was used as a control. The results were analyzed according to the CLSI breakpoints. The 17 antibacterial agents tested consisted of the macrolide azithromycin (AZM), the quinolone nalidixic acid (NAL), the fluoroquinolone ciprofloxacin (CIP), the aminoglycosides amikacin (AMI) and streptomycin (STR), the phenylpropanol chloramphenicol (CHL), the tetracyclines tetracycline (TET) and tigecycline (TIG), the folate pathway antagonist sulfamethoxazole (SMX); the lipopeptide polymyxin (PMX), and the β-lactams, ampicillin (AMP), ampicillin/sulbactam (AMS), ceftazidime (CAZ), ceftazidime/avibactam (CZA), cefotaxime (CTX), ertapenem (ETP), and meropenem (MEM).

## Whole-genome sequencing and assembly

The genomes of *S. enteritidis* strains were extracted using a bacterial genomic DNA extraction kit and sequenced using a NextSeq 550 sequencer (Illumina, San Diego, CA, USA). Quality control analysis of raw genomic sequencing data was performed with FastQC (v0.11.9) and fastp (v0.23.2) was used to remove low-quality data. Assembly was performed with SPAdes_3. 14.1 (https://github.com/ablab/spades) after removal of fragments with < 1000 bp of overlap. Information such as genome size and GC content were obtained using RAST (https://rast. nmpdr.org/rast.cgi). Whole-genome sequencing was performed by Shanghai Berger Medical Technology Co., Ltd. (Shanghai, China).

## Multilocus sequence typing

The results of whole-genome sequencing were typed for site sequence classification by multilocus sequence typing (MLST) with mlst v2.23.0 (https://github.com/tseemann/mlst). The numbers of housekeeping genes (*aroC*, *dnaN*, *hemD*, *hisD*, *purE*, *sucA*, and *thrA*) were obtained by alignment with the original allele sequences in the database and the corresponding sequence type (ST) was determined by combining them.

## Analysis of drug resistance genes and virulence genes

Based on the results of whole-genome sequencing analysis, the CARD (https://card.mcmaster. ca/) and VFDB (http://www.mgc.ac.cn/VFs/) databases were used to predict the resistance genes and virulence genes using the parameters identity > 90.0% and coverage > 90.0%. Finally, comparative analysis was performed on genotype and drug resistance phenotypes.

## Genetic correlation analysis

*S. enterica* Typhi CT18 (GenBank accession no. NC_003198) was used as the reference genome. The core single nucleotide polymorphism (SNP) data set of strains in this study was constructed using KSNP3.01. Sequence alignment and homology analysis were performed on 43 *S. enteritidis* strains using KSNP3.01 with 1000 bootstrap replicates. Phylogenetic trees were constructed using the maximum likelihood method.

## Nucleotide sequence accession numbers

The sequences obtained in this study have been deposited in GenBank with accession numbers JBAKMS000000000–JBAKOH000000000 and JBAOJF000000000.

## Results

### Genome sequencing

The total genome length of 43 strains of *S. enteritidis* was 4,417,739–4,757,703 bp and the average GC content was 52.07–52.42% (S2 Table), consistent with the genomic characteristics of *S. enteritidis*.

### MLST molecular typing

MLST analysis based on the whole genome sequence showed that all 43 *S. enteritidis* isolates belonged to ST11 type (S3 Table), and the seven housekeeping genes were *aroC*, *dnaN*, *hemD*, *hisD*, *purE*, *sucA*, and *thrA*.

### Detection of AST and antimicrobial resistance genes

The 43 strains of *S. enteritidis* showed varying degrees of resistance to 17 antibiotics with resistance rates ranging from 2.33% to 100%. The highest resistance rate was observed for NAL (100%) followed by AMP and AMS (both 90.70%) (Table 1). In all, 35 of the 43 *S. enteritidis* strains were resistant to three or more antibacterial drugs, representing a total multidrug resistance rate of 81.40% (35/43). The 43 *S. enteritidis* strains showed 19 drug resistance spectra, among which AMP-AMS-NAL-STR-PMX and AMP-AMS-NAL-STR-TET-PMX, seen in eight *S. enteritidis* strains each, were predominant. *S. enteritidis* resistant to five antibiotics accounted for the greatest proportion among multidrug-resistant strains (31.43%, 11/35). The most resistant strain was detected in 2023 (S2023828), which was resistant to all 17 drugs examined (Table 2).

All 43 strains of *S. enteritidis* carried drug resistance genes, and a total of 23 drug resistance genes, were predicted by the analysis. Among these, 14 resistance genes were common to all 43 strains, i.e., *bacA*, *mdtK*, *marA*, *AAC(6′)-Iy*, *cpxA*, *golS*, *mdsA*, *mdsB*, *mdsC*, *sdiA*, *acrB*, *emrR*, *H-NS*, and *CRP* (Fig 1). The seven most common antimicrobial resistance genes were *marA*,

**Table 1. Antimicrobial susceptibility testing of 43 *Salmonella enteritidis* strains with 17 antimicrobial agents.**

| antimicrobial | sensitive (n,%) | intermediate (n,%) | resistant (n,%) |
|---|---|---|---|
| AMP | 4(9.30) | 0(0.00) | 39(90.70) |
| AMS | 4(9.30) | 12(27.91) | 27(62.80) |
| CAZ | 40(90.2) | 0(0.00) | 3(6.98) |
| CZA | 41(95.35) | 0(0.00) | 2(4.65) |
| CTX | 36(83.72) | 37(25.34) | 7(16.28) |
| ETP | 42(97.67) | 0(0.00) | 1(2.33) |
| MEM | 42(97.67) | 0(0.00) | 1(2.33) |
| NAL | 0(0.00) | 0(0.00) | 43(100) |
| CIP | 42(97.67) | 0(0.00) | 1(2.33) |
| AZM | 39(90.70) | 0(0.00) | 4(9.30) |
| AMI | 42(97.67) | 0(0.00) | 1(2.33) |
| STR | 10(23.26) | 6(13.95) | 27(62.79) |
| TET | 27(62.80) | 0(0.00) | 16(37.21) |
| TIG | 42(97.67) | 0(0.00) | 1(2.33) |
| CHL | 39(90.70) | 0(0.00) | 4(9.30) |
| SXT | 38(88.37) | 0(0.00) | 5(11.63) |
| PMX | 12(27.91) | 0(0.00) | 31(72.09) |

**Table 2. Drug resistance spectra of 43 *Salmonella enteritidis* strains to 17 antimicrobial agents.**

| Antibiotic resistant species (species) | Drug-resistant spectrum | Number of isolate (n) | The percentage (%) |
|---|---|---|---|
| 1 | NAL | 3 | 6.98 |
| 2 | AMP-NAL | 4 | 9.30 |
|  | NAL-STR | 1 | 2.33 |
| 3 | AMP-NAL-PMX | 4 | 9.30 |
| 4 | AMP-NAL-AMS-PMX | 2 | 4.65 |
|  | AMP-AMS-NAL-STR | 2 | 4.65 |
|  | AMP-NAL-STR-PMX | 1 | 2.33 |
|  | AMP-NAL-SXT-PMX | 1 | 2.33 |
| 5 | AMP-AMS-NAL-STR-TET | 1 | 2.33 |
|  | AMP-AMS-NAL-STR-PMX | 8 | 18.60 |
|  | AMP-AMS-NAL-CTX-TET | 1 | 2.33 |
|  | AMP-CAZ-CTX-NAL-PMX | 1 | 2.33 |
| 6 | AMP-AMS-NAL-STR-TET-PMX | 8 | 18.60 |
|  | AMP-AMS-NAL-STR-SXT | 1 | 2.33 |
|  | AMP-AMS-CTX-NAL-TET-PMX | 1 | 2.33 |
| 9 | AMP-AMS-CTX-NAL-AZM-STR-TET-CHL-PMX | 1 | 2.33 |
|  | AMP-AMS-CTX-NAL-AZM-STR-TET-CHL-SXT | 1 | 2.33 |
| 12 | AMP-AMS-CAZ-CZA-CTX-NAL-AZM-STR-TET-CHL-SXT-PMX | 1 | 2.33 |
| 17 | AMP-AMS-CAZ-CZA-CTX-ETP-MEM-NAL-CIP-AZM-AMI-STR-TET-TIG-CHL-SXT-PMX | 1 | 2.33 |

*sdiA*, *acrB*, *golS*, *mdsA*, *mdsB*, and *mdsC*, of which *marA* was detected in the greatest number of strains (Table 3).

## Virulence genes

Analysis using the Virulence Factor Database (VFDB) showed that the 43 *S. enteritidis* strains possessed a total of 108 virulence genes and had 20 pathogenic mechanisms. Among them, 96 virulence genes were shared by the 43 strains, and the type III secretion system was the main pathogenic mechanism (Fig 2). (S4 Table and S5 Table)

## Genetic correlation analysis

A maximum likelihood tree based on core SNPs was constructed using *S. enterica* Typhi CT18 (GenBank accession no. NC_003198) as the reference genome. The 43 strains of *S. enteritidis* were divided into three clusters, with cluster 1 containing 10 strains, cluster 2 containing 11 strains, and cluster 3 containing 22 strains. Two strains of *S. enteritidis* originating from food were located within cluster 3, and the 10 strains from concentrated outbreaks formed independent branches all of which were also within cluster 3 (Fig 3).

## Discussion

*S. enteritis* can cause severe invasive infections [29]. The main clinical manifestation is self-limited acute enteritis, but it can also develop into invasive infections outside the intestine and cause systemic disease [30]. Therefore, it is important to monitor drug resistance and analyze the genetic characteristics of *S. enteritidis* to not only provide a scientific basis for clinical treatment but also to support tracing of the sources of infection and establish a local *Salmonella* database.

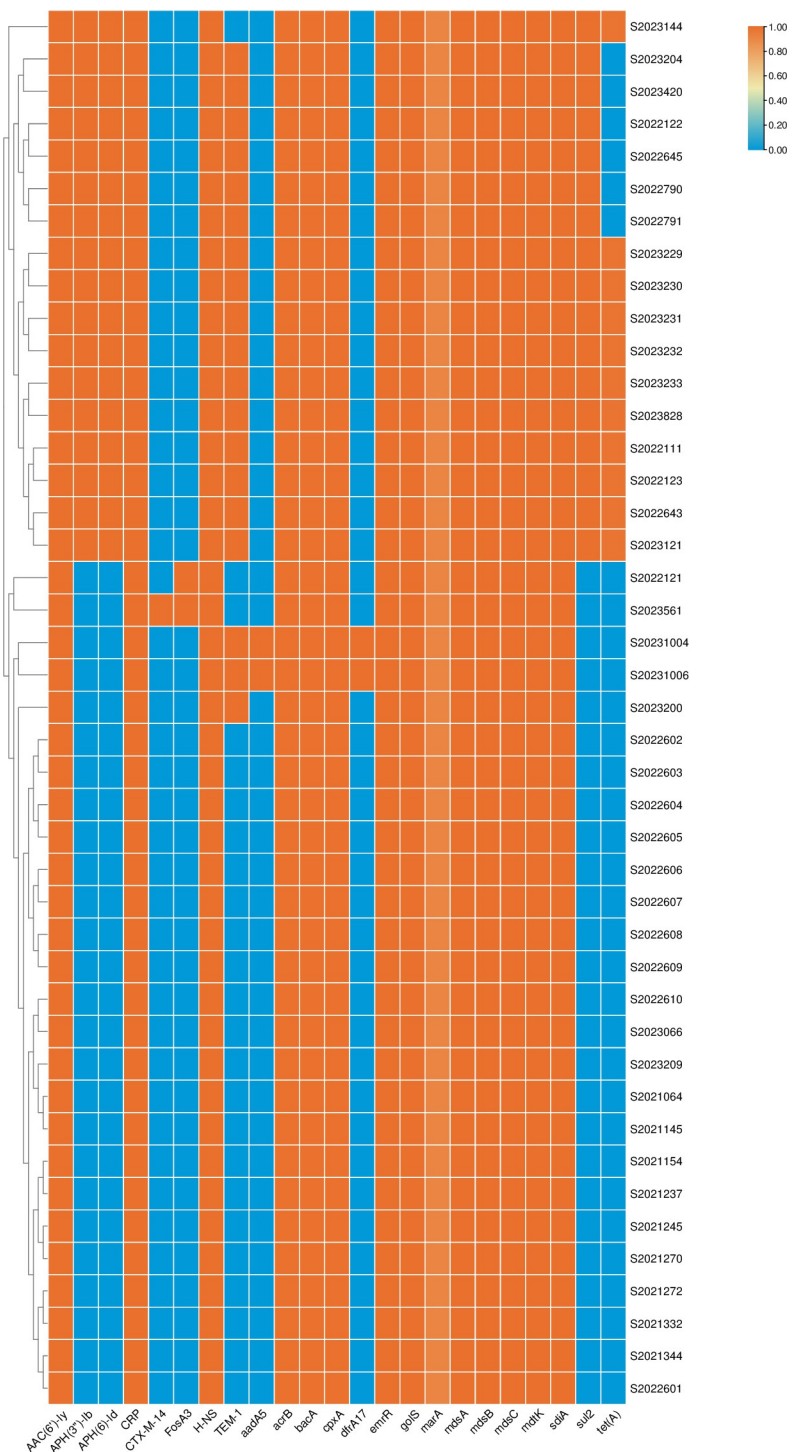

**Fig 1. Drug resistance genes detected in the 43 *S. enteritidis* isolates examined in this study.**

Cephalosporins and quinolones are important antibiotics used to treat *S. enteritidis* infections [31]. However, there have been reports that some strains have developed resistance to these antibiotics [32]. The results of drug sensitivity testing in *S. enteritidis* in the Huzhou area indicate varying degrees of resistance to all 17 antibiotics examined. High levels of resistance

**Table 3. Drug resistance genotyping and drug resistance of *Salmonella enteritidis* strains.**

| GENE | GENE_COUNT | RESISTANCE_COUNT | RESISTANCE |
|---|---|---|---|
| *bacA* | 43 | 1 | peptide |
| *mdtK* | 43 | 1 | fluoroquinolone |
| *marA* | 43 | 12 | carbapenem;cephalosporin;cephamycin;fluoroquinolone;glycylcycline;monobactam;penam;penem;phenicol; rifamycin;tetracycline;triclosan |
| *AAC(6')-Iy* | 43 | 1 | aminoglycoside |
| *cpxA* | 43 | 2 | aminocoumarin;aminoglycoside |
| *golS* | 43 | 7 | carbapenem;cephalosporin;cephamycin;monobactam;penam;penem;phenicol |
| *mdsA* | 43 | 7 | carbapenem;cephalosporin;cephamycin;monobactam;penam;penem;phenicol |
| *mdsB* | 43 | 7 | carbapenem;cephalosporin;cephamycin;monobactam;penam;penem;phenicol |
| *mdsC* | 43 | 7 | carbapenem;cephalosporin;cephamycin;monobactam;penam;penem;phenicol |
| *sdiA* | 43 | 8 | cephalosporin;fluoroquinolone;glycylcycline;penam;phenicol;rifamycin;tetracycline;triclosan |
| *acrB* | 43 | 8 | cephalosporin;fluoroquinolone;glycylcycline;penam;phenicol;rifamycin;tetracycline;triclosan |
| *emrR* | 43 | 1 | fluoroquinolone |
| *H-NS* | 43 | 6 | cephalosporin;cephamycin;fluoroquinolone;macrolide;penam;tetracycline |
| *CRP* | 43 | 3 | fluoroquinolone;macrolide;penam |
| *APH(6)-Id* | 28 | 1 | aminoglycoside |
| *APH(3")-Ib* | 28 | 1 | aminoglycoside |
| *TEM-1* | 26 | 4 | cephalosporin;monobactam;penam;penem |
| *sul2* | 17 | 1 | sulfonamide |
| *tet(A)* | 11 | 1 | tetracycline |
| *FosA3* | 2 | 1 | fosfomycin |
| *dfrA17* | 2 | 1 | diaminopyrimidine |
| *aadA5* | 2 | 1 | aminoglycoside |
| *CTX-M-14* | 1 | 1 | cephalosporin |

were observed to NAL, AMP, AMS, and STR, consistent with previous reports [33–35], indicating that these agents are no longer suitable for the treatment of *Salmonella* infection in Huzhou area. In addition, multiple drug resistance of *S. enteritidis* is becoming increasingly problematic in the Huzhou area with 81.40% of strains showing resistance to three or more

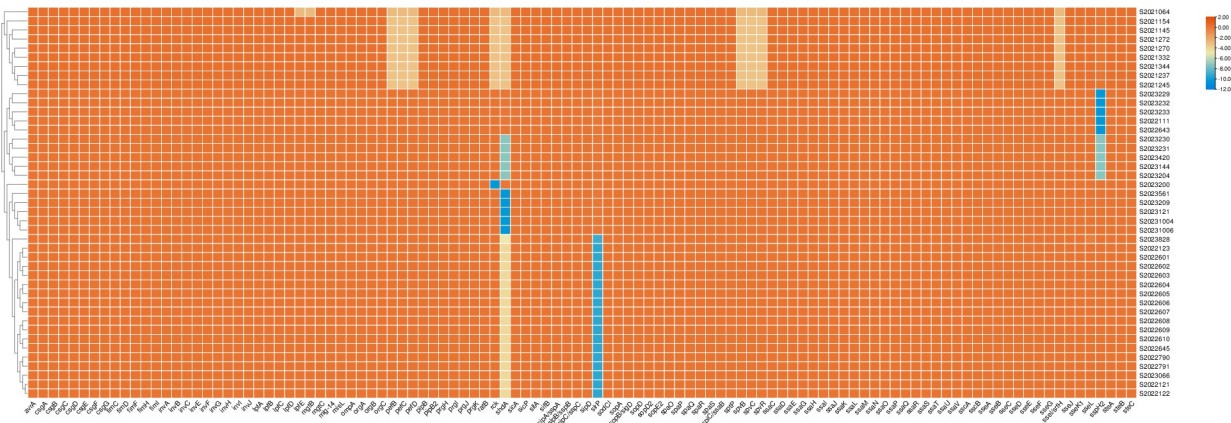

**Fig 2. Virulence factors detected in the 43 *S. enteritidis* isolates examined in this study.**

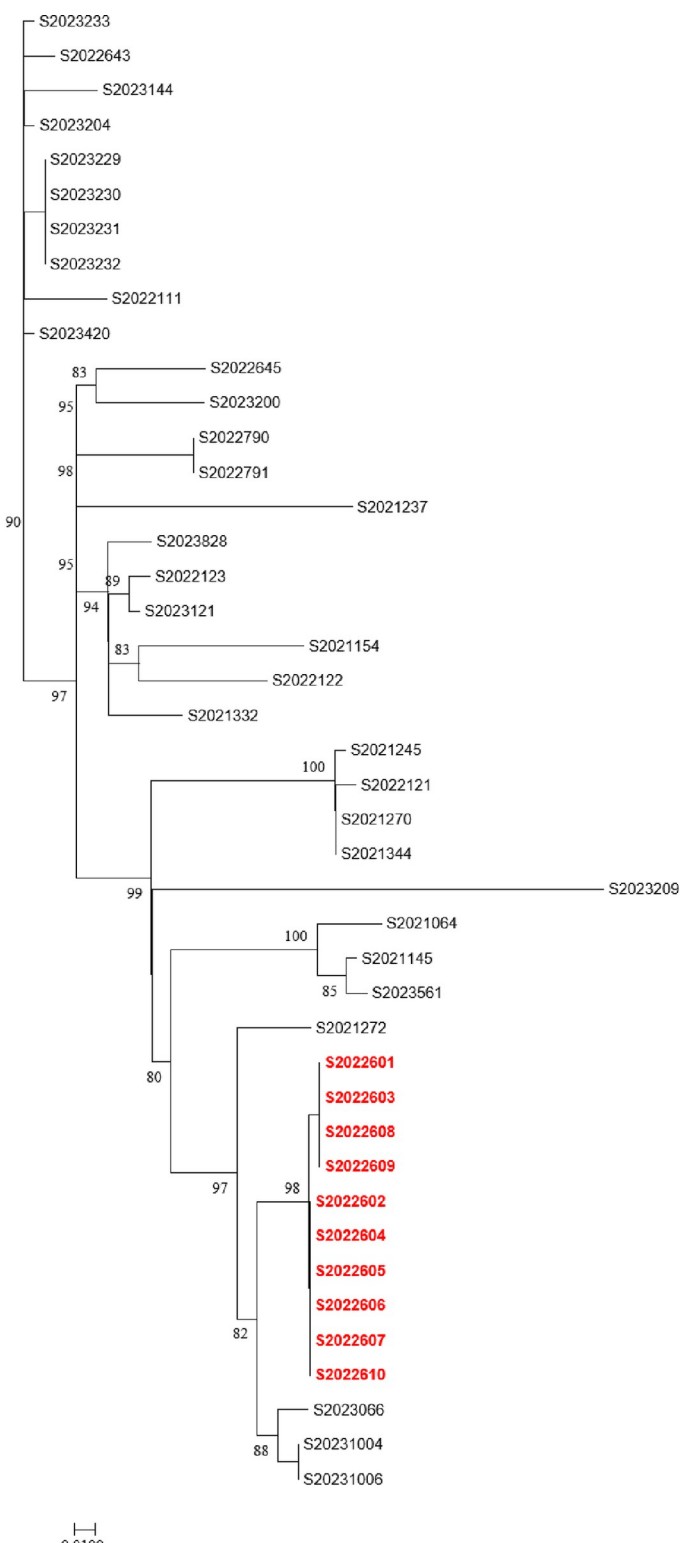

**Fig 3. Phylogenetic tree based on core genome SNPs of 43 *S. enteritidis* isolates.**

drugs, with marked resistance to NAL, AMP, and AMS. Twenty-six strains were resistant to five or more antibiotics, and the greatest spectrum of multiple drug resistance involved resistance to all 17 antibiotics. The emergence of multidrug resistance is likely due to the abuse of antibiotics and horizontal transmission of resistance genes. Therefore, it is necessary to improve prescription practices, reduce the occurrence of drug-resistant bacteria, and prevent the spread of drug-resistant strains.

The 43 strains were highly resistant to fluoroquinolones, β-lactams, and aminoglycosides, and their genotypes and phenotypes were essentially the same. Hyeon et al. [36] reported that some genes are usually not sufficient to cause changes in the resistance phenotype in *Salmonella* [37, 38]. Our results also show that the drug resistance genotype and phenotype were not consistent, for example, *sul2* was detected in 17 strains, while only 5 strains showed SMX resistance. In addition, four strains were resistant to CHL, but no related drug resistance genes were detected. This may have been due to other unknown genes, and further study is warranted.

With advances in bioinformatics techniques, WGS has been widely used to analyze pathogenic bacteria due to its high resolution, good repeatability, and rapid and high-throughput characteristics [39].

It is also becoming increasingly common to use bioinformatics to predict serotypes based on *Salmonella* whole-genome sequence data. In this study, 43 *S. enteritidis* strains isolated in Huzhou in 2021–2023 were subjected to whole-genome sequencing, and compared to strains in public databases to construct a phylogenetic tree based on SNPs. The results divided the 43 strains of *S. enteritidis* into three clusters. Ten strains from outbreaks were clustered in independent branches. In addition, all three clusters contained strains from different regions and/or isolation times, indicating that *S. enteritidis* in Huzhou was composed of sporadic strains.

MLST is a classical bacterial molecular typing technique widely used to identify relationships among bacterial clones [40]. In this study, MLST analysis showed that all 43 strains of *S. enteritidis* were ST11 type, and included seven housekeeping genes, consistent with a previous report [41, 42]. The results indicated close correlations between sporadic *S. enteritidis* in Huzhou. In addition, the detection rate of *S. enteritidis* virulence factors was high, with 96 (88.89%) virulence genes shared by the 43 strains of *S. enteritidis* examined in this study.

Some studies have reported a positive correlation between virulence genes and the pathogenicity of *Salmonella* [43, 44]. The *Salmonella* plasmid virulence (*spv*) genes are commonly present on virulence plasmids of many *Salmonella* species, including *spvRABCDE*, which mainly mediates cytotoxicity during macrophage differentiation and apoptosis [45]. In this study, 34 strains of *S. enteritidis* were shown to carry *spvRBC*, suggesting a relatively high toxic plasmid carrying rate of *S. enteritidis* in Huzhou.

The major limitations of this study were that we analyzed only *S. enteritidis* and the number of strains was small. Further studies are required with a greater detection range and quantity of samples to evaluate the effects of *S. enteritidis*-related diseases and provide a scientific basis for the prevention and control of *S. enteritidis* infection in Huzhou.

## Conclusion

We analyzed the characteristics of AST and WGS in *S. enteritidis* strains isolated from patients with diarrhea in Huzhou, China. All isolates were of the ST11 type, which were sporadic strains. The carrying rate of virulence genes was high, and drug resistance was serious, with many multidrug-resistant strains. Close attention should be paid to the emergence of drug-resistant strains and management of antibiotics should be strengthened. Our results provide a

background for the prevention and treatment of foodborne diseases caused by *S. enteritidis* in Huzhou, Zhejiang Province, China.

## Supporting information

**S1 Table. The information of 43 *Salmonella enteritidis* isolates strain.**
(DOCX)

**S2 Table. Genome assembly profile of *Salmonella enteritidis*.**
(DOCX)

**S3 Table. Results of MLST typing of *Salmonella enteritidis* genome.**
(DOCX)

**S4 Table. Genomic virulence gene results of *Salmonella enteritidis*.**
(DOCX)

**S5 Table. *Salmonella enteritidis* genome virulence gene pathogenic system.**
(DOCX)

## Acknowledgments

We thank the outpatients, nurses, and clinicians of the participating hospitals for their cooperation with the study.

## Author Contributions

**Data curation:** Wei Yan, Deshun Xu, Xiaofang Wu.

**Methodology:** Liping Chen.

**Software:** Deshun Xu, Liping Chen.

**Writing – original draft:** Wei Yan.

**Writing – review & editing:** Xiaofang Wu.

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
