## [Decision Letter · Decision Letter 0]

8 Apr 2024

PONE-D-24-09010Antimicrobial resistance and genome characteristics of Salmonella enteritidis from Huzhou,ChinaPLOS ONE

Dear Dr. Xu,

Thank you for submitting your manuscript to PLOS ONE. After careful consideration, we feel that it has merit but does not fully meet PLOS ONE’s publication criteria as it currently stands. Therefore, we invite you to submit a revised version of the manuscript that addresses the points raised during the review process.

We look forward to receiving your revised manuscript.

Kind regards,

Mabel Kamweli Aworh, DVM, MPH, PhD. FCVSN

Academic Editor

PLOS ONE

Journal Requirements:

Additional Editor Comments :

In addressing the reviewer's comments please address the following additional issues.

1. In your revision, please provide line numbers on the revised manuscript to make it easier for reviewers to review the revised manuscript.

2. Please engage the services of a native English speaker to revise the manuscript for typographical and grammatical errors. Delete any Chinese inscriptions in the manuscript, especially in the results section. Define any abbreviations upon first use in the manuscript.

3. Genes should be italicized and written in small letters all through the manuscript.

4. In discussing your results, do not repeat them; rather, provide a possible explanation for your findings while comparing them to the reports of other similar studies.

5. In the second paragraph of the discussion section, please italicize "*Escherichia coli"* and any other bacteria mentioned in the manuscript. Please italicize "*Salmonella*" all through the manuscript.

6. Highlight the main limitations of the current study in the last paragraph of the discussion section, please.

Reviewers' comments:

Reviewer's Responses to Questions

**Comments to the Author**

1. Is the manuscript technically sound, and do the data support the conclusions?

Reviewer #1: Yes

Reviewer #2: Partly

2. Has the statistical analysis been performed appropriately and rigorously? 

Reviewer #1: No

Reviewer #2: N/A

3. Have the authors made all data underlying the findings in their manuscript fully available?

Reviewer #1: Yes

Reviewer #2: Yes

4. Is the manuscript presented in an intelligible fashion and written in standard English?

Reviewer #1: No

Reviewer #2: No

5. Review Comments to the Author

Reviewer #1: The authors have a good study question but were not able to properly put it in writing. There is no justification for the study, the methodology was not properly referenced for reproducibility and there were several grammatical and writing errors. Also, all authors referenced in the article need to be fully stated in the reference section and not stating et al., in places where the full names are supposed to be written. Not to say that there were scientific names that were not italicized and confusing species of Salmonella in the result section. Authors need to proof-read, justify the document and pay attention to details.

Reviewer #2: General Comments:

Personally, I would like to see the letter of approval, especially since the author claims that the consent was given orally. The paper could benefit from a thorough revision, particularly in terms of punctuation. The lack of line numbers makes it challenging to pinpoint specific areas where punctuation is missing. Additionally, there are several acronyms that are not fully spelled out. I also noticed possible translation errors, as it appears the author may have translated the paper from another language to English. I recommend a careful review to ensure accurate translation. Finally, the author should pay much attention on their discussion because they did not discuss all results, and the conclusion is weak.

Abstract:

The authors have failed to state the purpose of their study, despite attempting to define the problem. This is a crucial element that could engage the reader’s interest in the article.

Introduction:

The sentence, “Multi-drug resistant bacteria and even super bacteria have emerged, which lead to the emergence of new epidemiological and clinical characteristics of the disease,” seems redundant and does not add value. It appears to repeat previously mentioned information. The preceding sentence could also be restructured and potentially supplemented with a citation.

Results:

It was disappointing to see that the authors did not reference the supplemental material, which, in my opinion, could have been effectively utilized in their results to strengthen their arguments.

Discussion:

The authors should consider using more direct language, as exemplified by their use of the vague term “some strains.” The discussion lacks depth and fails to highlight the practical implications of the results. There are no recommendations beyond drug resistance. Apparently, the genomic results are not discussed at all, and they applied genomic sequencing without a clear plan of how the results might be useful.

Conclusion:

The conclusion is weak. It merely restates the methods without summarizing the results or discussing their potential benefits to the community. It gives the impression that the authors were more interested in applying the methods and maintaining records than in the study’s purpose.

6. PLOS authors have the option to publish the peer review history of their article (what does this mean?). If published, this will include your full peer review and any attached files.

Reviewer #1: No

Reviewer #2: No

---

## [Author Response · Author response to Decision Letter 0]

22 Apr 2024

Dear  editors: Thank you very much for your letter and the comments from the referees about our paper submitted to PLOS ONE (PONE-D-24-09010).

 We have carefully addressed all of the comments from the reviewers, as outlined in the point-by-point responses attached below. We hope that you find the revised manuscript now acceptable for publication in PLOS ONE.

 The English in this document has been checked by at least two professional editors, both native speakers of English. For a certificate, please see:

http://www.textcheck.com/certificate/axviDO

Thanks for your attention

Yours Sincerely 

Deshun Xu

Responses to the editor’s comments: 

(Q: as comments, A: as our responses)

Q: Can authors please justify the article? 

Kindly proof read and ensure proper spacing after a comma or full stops. Pay attention to details! 

A: We had ensured proper spacing after a comma or full stops and revised other details in the manuscript. 

INTRODUCTION 

Q: Line 41 “Salmonella is a zoonotic pathogen. It is also an important pathogenic factor 42 causing foodborne diseases [2,3] and the main cause of bacterial food poisoning [4]. 43 Contaminated food and water source or infected individuals by Salmonella can 44 become the source of infection and spread to the outside, posing a serious threat to 45 human and animal health[1]”. 

Any reason for starting the referencing numbers with 2, 3 and 4 before “1”? If none please correct it accordingly. 

A: We have made corrections it accordingly in the manuscript.

Q: Line 48 – 49 “gastroenteritis type, septicemia type (typhoid type) and 49 local infection type” Kindly reference this statement 

A: We have added references “Mingfeng SH, Rongxing C, Kai ZH, Hongjian CH. Etiological and clinical characteristics of 92 children with Salmonella enteritis[J].Chin J Nosocomiol, 2022, 32(14): 2212-2217.” in the manuscript.

Q: Authors need to state clearly the justification for the study as dearth of information of the said topic in the region studied. 

A: We added justification for the study in the introduction: There have been no previous reports on the drug resistance and genetic characteristics of S. enteritidis in the Huzhou area。

METHODOLOGY 

Q: The authors need to cite references for the methodology “as described by ……” for reproducibility. 

A: We added 2 references in the manuscript: “ Bacteria were isolated according to the methods described in Diagnostic Criteria for Infectious Diarrhea (WS271-2007) and National Standards for Food Safety (GB4789.4-2016) ”.

Q: Lines 86 – 88 Ethics statement 

This study was approved by the human research ethics committee of the Huzhou Center for Disease Control and Prevention. 

kindly state the approval number 

A: We added the approval number(HZ2020007) in the manuscript.

Q: Line 151 “and the 7 housekeeping genes were aroC 152 (5), dnaN (2), hemD (3), hisD (7), purE(6), sucA (6) and thrA (11).” 

These values when added gives 40 as against 43 stated. Any reason for the discrepancy?Nothing was said about the statistical analysis in the methodology or none was carried out? 

A: The data in parentheses represent an identifier for every unique allele sequence, rather than the number of strains, and were used to determine the ST genotype of the strain. We removed these data so as not to cause any misunderstanding.

Statistical analysis is rarely used in this paper.

RESULT 

Q: Can authors confirm the species of Salmonella especially in the result because there are several confusing names? 

A: We have made careful modifications as required.

Q: If authors are not stating a specific species or simple sp. there is no need italicizing the Salmonella 

A: We have made careful modifications as required.

REFERENCES 

Q: While writing the authors of articles cited, kindly write out the names of all authors and desist from using et al., in the reference section. Check online for the names of all authors. If truly these articles were downloaded and read before citing them, the authors should have all the names of the authors of articles cited in this study. See lines 286, 289, 301, 310, 314, 317, 320, 324, 327, 330, 333, 338, 342, 348, 351, 356, 362, 365, 368,371, 374, 381, 387 

A: We have made careful modifications as required. The names of all authors have been written out in the reference section.

Q:5. Please include captions for your Supporting Information files at the end of your manuscript, and update any in-text citations to match accordingly. Please see our Supporting Information guidelines for more information: http://journals.plos.org/plosone/s/supporting-information.

A: Supplementary Material had been submitted at the time of initial submission of the paper.

Q: Please review your reference list to ensure that it is complete and correct. If you have cited papers that have been retracted, please include the rationale for doing so in the manuscript text, or remove these references and replace them with relevant current references. Any changes to the reference list should be mentioned in the rebuttal letter that accompanies your revised manuscript. If you need to cite a retracted article, indicate the article’s retracted status in the References list and also include a citation and full reference for the retraction notice.

A: We have carefully checked the references.

Additional Editor Comments :

In addressing the reviewer's comments please address the following additional issues.

Q: 1. In your revision, please provide line numbers on the revised manuscript to make it easier for reviewers to review the revised manuscript.

A: We have added line numbers to the manuscript as requested.

Q: 2. Please engage the services of a native English speaker to revise the manuscript for typographical and grammatical errors. Delete any Chinese inscriptions in the manuscript, especially in the results section. Define any abbreviations upon first use in the manuscript.

A: The English in this document has been checked by at least two professional editors, both native speakers of English. For a certificate, please see:

http://www.textcheck.com/certificate/axviDO

Q: 3. Genes should be italicized and written in small letters all through the manuscript.

A: We have made the changes as requested.

Q: 4. In discussing your results, do not repeat them; rather, provide a possible explanation for your findings while comparing them to the reports of other similar studies.

A: We removed the same content as the results from the discussion and added comparisons with similar studies to the discussion.

Q: 5. In the second paragraph of the discussion section, please italicize "Escherichia coli" and any other bacteria mentioned in the manuscript. Please italicize "Salmonella" all through the manuscript.

A: We have made the changes as requested.

Q: 6. Highlight the main limitations of the current study in the last paragraph of the discussion section, please.

A: The main limitations of the current study were added in the last paragraph of the discussion section.

5. Review Comments to the Author

Q: Reviewer #1: The authors have a good study question but were not able to properly put it in writing. There is no justification for the study, the methodology was not properly referenced for reproducibility and there were several grammatical and writing errors. Also, all authors referenced in the article need to be fully stated in the reference section and not stating et al., in places where the full names are supposed to be written. Not to say that there were scientific names that were not italicized and confusing species of Salmonella in the result section. Authors need to proof-read, justify the document and pay attention to details.

A: We have carefully revised the above content as required.

Reviewer #2: General Comments:

Q: Personally, I would like to see the letter of approval, especially since the author claims that the consent was given orally. The paper could benefit from a thorough revision, particularly in terms of punctuation. The lack of line numbers makes it challenging to pinpoint specific areas where punctuation is missing. Additionally, there are several acronyms that are not fully spelled out. I also noticed possible translation errors, as it appears the author may have translated the paper from another language to English. I recommend a careful review to ensure accurate translation. Finally, the author should pay much attention on their discussion because they did not discuss all results, and the conclusion is weak.

A: We had submitted the human research ethics committee of the Huzhou Center for Disease Control and Prevention(approval number: HZ2020007). The name of file is “approval document(English)”. And asked to modify the relevant content.

The English in this document has been checked by at least two professional editors, both native speakers of English. For a certificate, please see:

http://www.textcheck.com/certificate/axviDO

Abstract:

Q: The authors have failed to state the purpose of their study, despite attempting to define the problem. This is a crucial element that could engage the reader’s interest in the article.

 A: the purpose of study were added in abstract: This study investigated the drug resistance and genomic characteristics of S. enteritidis isolated from clinical and food sources in Huzhou, Zhejiang Province, China, from February 1 2021, to December 30, 2023.

Introduction: 

Q:The sentence, “Multi-drug resistant bacteria and even super bacteria have emerged, which lead to the emergence of new epidemiological and clinical characteristics of the disease,” seems redundant and does not add value. It appears to repeat previously mentioned information. The preceding sentence could also be restructured and potentially supplemented with a citation.

A: We removed “Multi-drug resistant bacteria and even super bacteria have emerged, which lead to the emergence of new epidemiological and clinical characteristics of the disease,” and the sentence was also rearranged.

Results: 

Q: It was disappointing to see that the authors did not reference the supplemental material, which, in my opinion, could have been effectively utilized in their results to strengthen their arguments.

A: Supplementary material file had been submitted at the time of initial submission of the paper.

Discussion:

Q: The authors should consider using more direct language, as exemplified by their use of the vague term “some strains.” The discussion lacks depth and fails to highlight the practical implications of the results. There are no recommendations beyond drug resistance. Apparently, the genomic results are not discussed at all, and they applied genomic sequencing without a clear plan of how the results might be useful.

A: By analyzing the relationship between drug resistance genes and drug resistance phenotypes, we found that 12 strains, including S2021237, carried sul2 gene, but were sensitive to trimethoprim-sulfamethoxazole. In addition, 4 strains, including S2023209, were resistant to chloramphenicol but no chloramphenicol resistance genes were found. At the same time, we removed the expression of ambiguous terms in the text.

Conclusion:

Q: The conclusion is weak. It merely restates the methods without summarizing the results or discussing their potential benefits to the community. It gives the impression that the authors were more interested in applying the methods and maintaining records than in the study’s purpose.

A: We reformatted the conclusion section.

---

## [Decision Letter · Decision Letter 1]

9 May 2024

PONE-D-24-09010R1Antimicrobial Resistance and Genome Characteristics of Salmonella enteritidis  from Huzhou, ChinaPLOS ONE

Dear Dr. Xu,

Thank you for submitting your manuscript to PLOS ONE. After careful consideration, we feel that it has merit but does not fully meet PLOS ONE’s publication criteria as it currently stands. Therefore, we invite you to submit a revised version of the manuscript that addresses the points raised during the review process.

We look forward to receiving your revised manuscript.

Kind regards,

Mabel Kamweli Aworh, DVM, MPH, PhD. FCVSN

Academic Editor

PLOS ONE

Journal Requirements:

Reviewers' comments:

Reviewer's Responses to Questions

**Comments to the Author**

1. If the authors have adequately addressed your comments raised in a previous round of review and you feel that this manuscript is now acceptable for publication, you may indicate that here to bypass the “Comments to the Author” section, enter your conflict of interest statement in the “Confidential to Editor” section, and submit your "Accept" recommendation.

Reviewer #1: All comments have been addressed

Reviewer #2: All comments have been addressed

2. Is the manuscript technically sound, and do the data support the conclusions?

Reviewer #1: Yes

Reviewer #2: Yes

3. Has the statistical analysis been performed appropriately and rigorously? 

Reviewer #1: N/A

Reviewer #2: N/A

4. Have the authors made all data underlying the findings in their manuscript fully available?

Reviewer #1: Yes

Reviewer #2: Yes

5. Is the manuscript presented in an intelligible fashion and written in standard English?

Reviewer #1: Yes

Reviewer #2: Yes

6. Review Comments to the Author

Reviewer #1: Authors have responded to all questions raised earlier and the manuscript can be accepted for publication.

Reviewer #2: I appreciate the current revised version, as it addresses most of the problems stated in the first version. I would also recommend it for publication.

A. However, the authors made a strong statement in lines 75 and 76: "There have been no previous reports on drug resistance and genetic characteristics of S enteriditis in the Huzhou area." Unless the authors are 100% sure there is none, I suggest revising this statement.

Based on a few searches, I realized that some co-authors have indeed contributed to previous works that include drug resistance in the Huzhou area. Nevertheless, these studies did not specifically investigate the genetic characteristics of S. enteriditis.

See articles:

1. Prevalence and Serotyping of Salmonella in Retail Food in Huzhou, China (D. Xu et al., 2024)

2. Characterization of Clinical Salmonella entericas Strains in Huzhou, China (D. Xu et al., 2022)

B. Furthermore, a review (though it is not a big issue) of the known genetic characteristics of S. enteriditis in other areas of China would be appreciated.

Example articles:

1. The Resistance and Virulence Characteristics of Salmonella Enteritidis Strain Isolated from Patients with Food Poisoning Based on the Whole-Genome Sequencing and Quantitative Proteomic Analysis (B. Xu et al., 2023)

2. Isolation, Identification, Antimicrobial Resistance, Genotyping, and Whole-Genome Sequencing Analysis of Salmonella Enteritidis Isolated from a Food-Poisoning Incident (Hou et al., 2024)

Bibliography:

Hou, Z., Xu, B., Liu, L., Yan, R., & Zhang, J. (2024). Isolation, Identification, Antimicrobial Resistance, Genotyping, and Whole-Genome Sequencing Analysis of Salmonella Enteritidis Isolated from a Food.Poisoning Incident. Polish Journal of Microbiology, 73(1), 69–89. https://doi.org/10.33073/PJM-2024-008

Xu, B., Hou, Z., Liu, L., Yan, R., Zhang, J., Wei, J., Du, M., Xuan, Y., Fan, L., & Li, Z. (2023). The Resistance and Virulence Characteristics of *Salmonella* Enteritidis Strain Isolated from Patients with Food Poisoning Based on the Whole-Genome Sequencing and Quantitative Proteomic Analysis. Infection and Drug Resistance, 16, 6567–6586. https://doi.org/10.2147/IDR.S411125

Xu, D., Chen, L., Lu, Z., & Wu, X. (2024). Prevalence and Serotyping of Salmonella in Retail Food in Huzhou China. Journal of Food Protection, 87(2), 100219. https://doi.org/10.1016/J.JFP.2024.100219

Xu, D., Ji, L., Yan, W., & Chen, L. (2022). Characterization of Clinical Salmonella entericas Trains in Huzhou, China. Canadian Journal of Infectious Diseases and Medical Microbiology, 2022. https://doi.org/10.1155/2022/7280376

7. PLOS authors have the option to publish the peer review history of their article (what does this mean?). If published, this will include your full peer review and any attached files.

Reviewer #1: No

Reviewer #2: No

---

## [Author Response · Author response to Decision Letter 1]

13 May 2024

Dear  editors: Thank you very much for your letter and the comments from the referees about our paper submitted to PLOS ONE (PONE-D-24-09010).

 We have carefully addressed all of the comments from the reviewers, as outlined in the point-by-point responses attached below. We hope that you find the revised manuscript now acceptable for publication in PLOS ONE.

Thanks for your attention

Yours Sincerely 

Deshun Xu

Responses to the editor’s comments: 

(Q: as comments, A: as our responses)

Q: However, the authors made a strong statement in lines 75 and 76: "There have been no previous reports on drug resistance and genetic characteristics of S enteriditis in the Huzhou area." Unless the authors are 100% sure there is none, I suggest revising this statement.

A: We have put "There have been no previous reports on drug resistance and genetic characteristics of S enteriditis in the Huzhou area. "changed to “As far as we know, there have been few previous reports on the drug resistance and genetic characteristics of S.enteritidis in the Huzhou area”.

Q: Based on a few searches, I realized that some co-authors have indeed contributed to previous works that include drug resistance in the Huzhou area. Nevertheless, these studies did not specifically investigate the genetic characteristics of S. enteriditis.

See articles:

1. Prevalence and Serotyping of Salmonella in Retail Food in Huzhou, China (D. Xu et al., 2024)

2. Characterization of Clinical Salmonella entericas Strains in Huzhou, China (D. Xu et al., 2022)

A: These two manuscripts are previous research conducted by our research group. One manuscript focuses on the Prevalence and Serotyping of Salmonella in Retail Food in Huzhou, while the other paper examines the Characterization of Clinical Salmonella entericas Strains in Huzhou. However, it is important to note that these studies did not specifically investigate the genetic characteristics of S. enteriditis.

Q: B. Furthermore, a review (though it is not a big issue) of the known genetic characteristics of S. enteriditis in other areas of China would be appreciated.

Example articles:

1. The Resistance and Virulence Characteristics of Salmonella Enteritidis Strain Isolated from Patients with Food Poisoning Based on the Whole-Genome Sequencing and Quantitative Proteomic Analysis (B. Xu et al., 2023)

2. Isolation, Identification, Antimicrobial Resistance, Genotyping, and Whole-Genome Sequencing Analysis of Salmonella Enteritidis Isolated from a Food-Poisoning Incident (Hou et al., 2024)

A: These two manuscripts have been cited in the article, with References 35 and 42 respectively.

---

## [Editor Report · Decision Letter 2]

15 May 2024

Antimicrobial Resistance and Genome Characteristics of Salmonella enteritidis  from Huzhou, China

PONE-D-24-09010R2

Dear Dr. Xu,

We’re pleased to inform you that your manuscript has been judged scientifically suitable for publication and will be formally accepted for publication once it meets all outstanding technical requirements.

Kind regards,

Mabel Kamweli Aworh, DVM, MPH, PhD. FCVSN

Academic Editor

PLOS ONE

Additional Editor Comments (optional):

All reviewer's comments have been addressed.
---

## [Editor Report · Acceptance letter]

22 May 2024

PONE-D-24-09010R2 

PLOS ONE

Dear Dr. Xu, 

I'm pleased to inform you that your manuscript has been deemed suitable for publication in PLOS ONE. Congratulations! Your manuscript is now being handed over to our production team.

Kind regards, 

on behalf of

Dr. Mabel Kamweli Aworh 

Academic Editor

PLOS ONE